# Mechanical Properties of Dental Alloys According to Manufacturing Process

**DOI:** 10.3390/ma14123367

**Published:** 2021-06-17

**Authors:** Ji-Min Yu, Seen-Young Kang, Jun-Seok Lee, Ho-Sang Jeong, Seung-Youl Lee

**Affiliations:** Medical Device Research Division, National Institute of Food and Drug Safety Evaluation, Cheongju-si 28159, Korea; yjm9223@korea.kr (J.-M.Y.); seenyoung@korea.kr (S.-Y.K.); junseok1025@korea.kr (J.-S.L.); hosa33@korea.kr (H.-S.J.)

**Keywords:** casting, dental alloy, dentistry, mechanical properties, milling, 3D printing

## Abstract

The purpose of this study is to investigate the effect of the fabrication method of dental prosthesis on the mechanical properties. Casting was produced using the lost wax casting method, and milling was designed using a CAD/CAM program. The 3D printing method used the SLS technique to create a three-dimensional structure by sintering metal powder with a laser. When making the specimen, the specimen was oriented at 0, 30, 60, and 90 degrees. All test specimens complied with the requirements of the international standard ISO 22674 for dental alloys. Tensile strength was measured for yield strength, modulus of elasticity and elongation by applying a load until fracture of the specimen at a crosshead speed of 1.5 ± 0.5 mm/min (*n* = 6, modulus of elasticity *n* = 3). After the tensile test, the cross section of the fractured specimen was observed with a scanning electron microscope, and the statistics of the data were analyzed with a statistical program SPSS (IBM Corp. Released 2020. IBM SPSS Statistics for Windows, Version 27.0. Armonk, NY, USA: IBM Corp.) and using Anova and multiple comparison post-tests (scheffe method). The yield strength was the highest at 1042 MPa at an angle of 0 degrees in the specimen produced by 3D printing method, and the elongation was the highest at 14% at an angle of 90 degrees in the specimen produced by 3D printing method. The modulus of elasticity was the highest at 235 GPa in the milled specimen. In particular, the 3D printing group showed a difference in yield strength and elongation according to the build direction. The introduction of various advanced technologies and digital equipment is expected to bring high prospects for the growth of the dental market.

## 1. Introduction

For the fabrication of dental prostheses, precision casting has long been used. Casting is another term called a wax loss method or an investment, and the method is as follows. Attach a sprue pin to the wax pattern made as a model of the specimen and put it in the investment material [1,2,3]. The sprue pin provides a passage for molten metal into the mold. After removing the wax pattern by heating the mold, when the cast body reaches the casting temperature, the molten metal is injected into the space, which is also called a wax loss method or an investment method. Casting technology is a traditional dental prosthesis manufacturing method that has been used for a long time, but since all processes are performed by hand, it takes a long time, and incomplete casting may occur; as a result, different outcomes may be obtained depending on the skill of the technician [1,2,3,4,5,6,7].

However, in recent years, as the manufacturing technology of dental prosthesis is further developed and digital technology, such as CAD (computer-aided design)/CAM (computer-aided manufacturing), is widely introduced in the dental field, many changes have occurred in the manufacturing method of dental prosthesis. In particular, an oral scanner is used in the process of taking an impression, and CAD/CAM equipment has already been commercialized and used in the process of making a model and a wax pattern [6,7,8,9].

The CAD/CAM system is a technology that produces a prosthesis in 3D form using a CAD program, saves it as an STL (stereolithography) file, and then manufactures the prosthesis in the form designed with various CAM equipment. Milling and 3D-printing methods using CAD/CAM systems are the technologies that have been introduced and actively used in the dental field because of their advantages. It is easier to manufacture than traditional casting methods and can more accurately control the appearance of the result of work [1,2].

The milling process solves the problem of porosity, which is a casting defect, in that the CAM process shortens the manufacturing time of the prosthesis and improves the precision of the framework. The CAM process uses a milling machine or computer numerical control (CNC) machining to cut the material with a machine, and all steps are controlled by a computer program [6,7,8]. However, the milling method consumes a lot of raw materials for the metal block and, in the case of using a high-strength metal material, it has the disadvantage of frequent wear of the milling tool and restricts detailed work [1,2,8,9].

The selective laser sintering (SLS) method, using 3D printing, is a metal additive manufacturing (AM) process in which metal powders are melted layer by layer using a high-energy laser beam according to a computer design to create a three-dimensional model. The 3D printed specimen has a disadvantage in that the manufacturing cost is not economical and has a brittle property. However, this method has a fast production speed, less material waste, and a significantly lower incidence of porosity [6,7,10,11,12,13,14,15].

In general, there are noble metal alloys and non-precious metal alloys as metal materials for manufacturing dental prosthesis. Pure gold or Au alloy is used as a direct filling material, casting material, wire rod, etc. However, since pure gold is too expensive and has high ductility, it cannot sufficiently withstand the occlusal force, so it is mainly used in the form of an alloy to increase mechanical strength and lower the price. Co–Cr alloys are non-precious metal alloys that are developed as a substitute for gold alloys, and have good mechanical properties, corrosion resistance, and excellent biocompatibility, so they are actively used as dental materials [16,17]. In particular, Co–Cr alloy was used as a metal material in this study because it has excellent mechanical properties, as well as corrosion resistance comparable to that of precious metal alloys in the oral cavity [17,18].

The mechanical properties requirements set out by international standards must be observed [7,19]. In order to find out how various prosthesis fabrication methods affect mechanical properties, a mechanical property test guided by an international standard for dental alloys, was performed. The international standard categorizes dental metal materials into six types based on 0.2% permanent set yield strength, elongation, and modulus of elasticity, and for clinical use, the requirements set out in the international standard must be satisfied [20].

In this study, we would like to investigate the effect of different fabrication methods on the mechanical properties of dental prostheses by using casting techniques, milling using CAD/CAM systems, and 3D printing processing methods. In addition, the introduction of advanced technologies, such as CAD/CAM and digital equipment used in this study, is practical and very economical, so it is expected to bring high prospects for the growth of the dental market.

## 2. Materials and Methods

### 2.1. Specimen Production

In order to find out whether the manufacturing method of the prosthesis affects the mechanical properties, the method of manufacturing the specimen was divided into three groups, as shown in Figure 1.

All specimens were made of Co–Cr alloy, and the dimensions of the specimens were manufactured in the form of 54 dumbbell-shaped tensile specimens according to ISO 22674 (Figure 2). Specimens were drawn using AutoCAD software (Inventor ver. 2015, Autodesk, Inc., San Rafael, CA, USA) according to the dimensions of the in Figure 2, and then converted into .stl file format and sent to the machine.

The composition of alloys by group is shown in Table 1.

The manufacturing process of the casting specimen is as follows. Dumbbell-shaped lead type waxes were produced using dental wax for casting. After the lead mold was produced, the injection line pin was attached to the thickest part of the mold and was then embedded in an investment material made by mixing phosphate-based powder and water. To completely remove the lead, the mold was heated in an oven. When the mold reached the casting temperature, the alloy was melted in using casting machines and injected into the mold space. The mold was cooled after casting, and then the investment material was removed, the injection line was cut, and the surface was oxidized. The casted specimens were cooled to room temperature and cleaned by sandblasting with alumina particles (50 μm).

A CAD/CAM program (Hyperdent CAM Software, v.8.2, 2018, Pistis, Incheon, Korea) was used in the milling process. In the CAM process, a dumbbell-shaped specimen was designed according to the drawing in Figure 2 using 3D data, and an STL file was created and then transferred to a milling machine (PM5-All, Pistis, Incheon, Korea) to delete the metal block for processing. The milled specimen was cleaned by sandblasting with alumina particles (50 μm).

Specimens prepared using a 3D printer (METALSYS 120D, Winforsys, Yongin-si, Korea) were printed by selective laser sintering (SLS). The STL file of the specimen to be manufactured was created through the CAD/CAM program used for milling, and then sent to a 3D printer (METALSYS 120D, Winforsys, Yongin-si, Korea) and manufactured by the SLS method. In the SLS method, a laser is selectively irradiated on the metal powder applied to the bed, and the metal powder is sintered with a laser to form a three-dimensional object. The metal powder material (particle size: 1–250 μm) was irradiated with a laser at a power of 130 W and a speed of 800 mm/s; it was then solidified into a thin layer of 20 μm and was repeatedly stacked. The metal powder was repeatedly laminated onto the solidified layer to complete the result. When preparing a specimen by laminating metal powders, as shown in Figure 3, directionality was arbitrarily given to each angle at 0 degrees (tensile direction and horizontal), 30 degrees, 60 degrees, and 90 degrees (vertical to tensile direction). All of the 3D printing specimen was cleaned by sandblasting with alumina particles (50 μm).

### 2.2. Mechanical Properties Test

Through the tensile test (specimen *n* = 6), 0.2% yield strength, tensile strength, and elongation were measured. For the tensile test, the gauge length of the test specimen was accurately measured within the gauge length interval to 0.01 mm using micrometers (MHT-2, Matsuzawa Seiki Co., Ltd., Akita-shi, Japan) or Vernier calipers (Mitutoyo, Co., Kanagawa, Japan). A tensile load was applied to the test specimen using a universal material tester (Instron 3367, Instron Co., Norfolk, MA, USA) at a cross head speed of 1.5 ± 0.5 mm/min until the specimen broke. This test was performed according to the Ref. [20] mechanical test procedure. Figure 4 is an example of performing a mechanical test. The elongation of the specimen was measured using an extensometer to obtain a continuous record without being affected by the tester compliance. The broken specimen was examined visually without a magnifying glass to determine whether the visible defects inside and outside the specimen were related to the damage of the specimen, in addition to whether the fractures occurred within the mark or the inscription line indicating the gauge length. If visible defects were observed or damage occurred outside the gauge length, the specimens and their results were rejected. The broken pieces were reattached and accurately measured to 0.02 mm with a portable microscope.

#### Modulus of Elasticity Using Tensile Strain Method

The modulus of elasticity was measured using the following Ref. [20] tensile strain method. We applied a strength equivalent to 60% of the 0.2% yield strength to the specimen, recorded the force and elongation, and then lowered the force to 5% of the 0.2% yield strength. After repeating this procedure at least 4 times, we calculated the elastic modulus according to the elastic modulus equation below (test specimen *n* = 3). We then measured two more specimens and recorded the average of the three most consistent values.
(1)$$E=\frac{\Delta P}{A}\cdot \frac{L}{\Delta L}$$
where E is the elastic modulus (in GPa), Δ*P* is the change in the force (in N), *A* is the cross-sectional area (in mm^2^), *L* is the initial gauge length (in mm), and Δ*L* is the extension (in mm).

### 2.3. Observation of Fracture Surface and Statistical Analysis

After the tensile test, the cross section of the test specimen was observed at ×100, ×200, ×500 and ×1000 magnification, using a scanning electron microscope (SNE 4500M, Scanning electron microscope, SEM, SEC Co., Ltd., Yongin-si, Korea).

Data were analyzed with a statistical program (IBM SPSS Statistics 27; IBM SPSS Inc., Armonk, NY, USA). The homogeneity of variance was verified using one-way ANOVA (analysis of variation) analysis. In order to identify significant differences between groups, post-mortem analysis was performed by applying a significance level (α = 0.05) adjusted by the Scheffe method in the post-analysis.

## 3. Results

Through the tensile test, the 0.2% yield strength, tensile strength, and elongation measurement results for each group can be checked in Table 2. The 0.2% Yield strength was highest in 3D printing results at 1008 ± 59 MPa, followed by 501 ± 33 MPa in casting and 438 ± 15 MPa in milling. The elongation was 11 ± 3% in casting, 12 ± 5% in milling, and 9 ± 2 in 3D printing, which showed the lowest value in elongation. The yield strength was higher in the horizontal direction compared to in the vertical build direction in the 3D printing groups (0°, 30°, 60°, 90°). The elongation tended to be inversely proportional to the yield strength of 5 ± 1% at 0 degrees, 6 ± 1% at 30 degrees, 9 ± 2% at 60 degrees, and 14 ± 2% at 90 degrees.

There was no significant difference between casting and milling in the overall average value of the yield strength of the three groups (*p* = 0.593), but there was a statistical difference between casting-3D printing and milling-3D printing (*p* < 0.001). In the 3D printing group by angle (0°, 30°, 60°, 90°), there was no significant difference (*p* = 0.195). When it comes to elongation, there was a statistical difference in casting-3D 0°, milling-3D 0°, and milling-3D 30°(*p* < 0.001). Additionally, the 3D-printing group showed a statistical difference in 3D 90°−3D 0° and 3D 90°-3D 30°(*p* < 0.001). As to the modulus of elasticity, the three did not show any significant difference within the group (*p* < 0.001).

Table 3 shows the 60% and 5% of the yield load for each specimen calculated at 0.2% yield strength. Tensile deformation tests were performed using the values of the measured yield strength. The modulus of elasticity measured by the tensile strain method was similar in casting group 226 GPa, milling group 235 GPa, 3D printing group 0 degree 219 GPa, 30 degree 212 GPa, 60 degree 206 GPa MPa, and 90 degree 198 GPa (*p* > 0.05).

Figure 5 and Figure 6 show scanning electron microscope (SEM) images of cross sections of the fractured portions of the specimens after the tensile strength test. After the tensile test, only specimens in which the fracture occurred within the gauge section were accepted. The observed microstructures of the fracture surfaces were slightly different, depending on the crystals produced during the fabrication of each specimen.

Figure 5a is the fracture surface of the specimen manufactured using the casting method and Figure 5b is the surface of the specimen by means of the milling method. Large and small pores were observed in the cross section of the casting specimen, and an uneven surface shape was observed in the cross section of the milling specimen as if the surface had been torn to one side.

Figure 6 is a cross section of a 3D-printing group specimen. Regardless of the build direction, the images of the fracture surfaces of the specimens produced by 3D printing were similar, and it was observed that the irregular crack surface and gaps were formed in various places of the fracture surface.

## 4. Discussion

When restoring a dental prosthesis, the enamel of natural teeth and the degree of wear of the prosthesis are factors that affect the chewing function of the oral cavity. Therefore, prostheses and natural teeth should have similar wear levels. In addition, the mechanical property test complies with Ref. [20] for dental alloys. According to the use of dental alloys, the relevant international standard is classified into six types, from type 0 to type 5, and based on mechanical properties, each type is classified according to 0.2% yield strength, elongation, and modulus of elasticity as shown in Table 4. The range of application is different for each type, and the highest type (Type 4 in Table 4) can include the low type (Type 0–4 in Table 4). The highest type can be applied from veneer crowns to removable partial dentures, clasps, and other parts which require high strength at the same time.

The results of mechanical properties (yield strength, elongation, and modulus of elasticity classified by the manufacturing method) in Table 2 and Table 3 showed similar trends in the casting and milling groups and, statistically, there was no significant difference between the groups (*p* > 0.05). The casting group had a yield strength of 501 MPa, an elongation of 11%, and a modulus of elasticity of 226 GPa, and the milling group had a yield strength of 438 MPa, an elongation of 12%, and a modulus of elasticity of 235 GPa. Mechanical properties were similar between the two groups. As shown in Table 4, it seems to be applicable to the type 4–5 alloy.

Figure 5 is an image of the fracture surface of the casting and milling group. In the specimen of the Casting group, several large and small pores were observed as a whole [7], and, in the milling group, an uneven surface with a tear in one direction was found. Despite the porosity that appeared in the process of casting [21], the yield strength was slightly higher in the casting group compared to the milling group, and this difference is expected to be affected by the chemical composition of the alloy.

Co-based alloys have different mechanical properties depending on the content of the constituent alloy [17,18]. The W, Mo, and Ni of the solid solution strengthening element can improve the strength depending on the content [22]. In the case of 3D-printed specimens, the yield strength was overwhelmingly superior to that of other groups, but it is difficult to judge that the mechanical properties are the highest because the elongation is low and brittle. The Co–Cr alloy used in the test is composed of various elements, such as Mo, W, Ni, and Si, and the content of each element is affected by strength and corrosion resistance. However, since the test specimen was manufactured in three ways, it is somewhat difficult to judge the mechanical properties based on the components alone. It is believed that the mechanical properties are affected by complex interactions, such as the components that make up the alloy and the way the prosthesis is manufactured. Therefore, if the specimen manufacturing process method and test conditions are the same, it is considered that the amount of the element constituting the alloy affects the mechanical properties.

In the case of the 3D-printing group produced by selectively melting metal powder with a laser, the yield strength was superior to that of the previous two groups. Unlike the casting technique, the SLS technique melts and solidifies the metal powder with a high-temperature laser; thus, unlike the casting technique, the incidence of pores is significantly lower [10,12,23]. That being said, the XRD measurement was not performed in this study, XRD data of cast specimens and 3D-printed specimens were obtained through previous research investigations. There was a tendency for the α phase to be more dominant in the 3D printing group than in the casting group [24]. Co–Cr alloy undergoes a phase transition to α phase face centered cubic(FCC) at high temperature and the ε hexagonal closest packed(HCP) phase at low temperature. In the 3D printing method of melting metal powder with a laser, α phase is predominant, and it is believed that the higher intensity was observed compared to the other two groups [21,22,23,24].

The yield strength of the 3D printing group showed a statistically significant difference when compared with the other two groups (Casting, milling) (*p* < 0.05). The yield strength of 3D printing was the highest at 1024 MPa in the specimen manufactured at an angle of 0 degrees. In particular, at 0°, the highest yield strength was recorded among all groups, but the elongation was the lowest. The 0-degree angle in Figure 3a was parallel to the direction in which 3D printing laminates the metal powder and the loading direction in the tensile test. During the manufacturing process of the 3D printing specimen, the metal powder melts and slight gaps, such as pores, may be formed between the laminated layers, which may cause defects. These gap defects are expected to cause fractures during the tensile test, elongate along the loading direction of the force, and then break [25]. When the stacking direction and loading direction are horizontal, as shown in Figure 3d, this sagging is considered to be shorter. From these results, it is believed that advantageous results can be produced using various variables such as the size of the powder, the speed of the laser, the specifications of the 3D-printing equipment, and the manufacturing angle. The introduction of various advanced technologies and digital equipment has high prospects for the growth of the dental market. Figure 6 is an image of the fracture surface of the 3D-printing group, and irregular cracks and gaps can be observed in various places. The anisotropy of the material is thought to have an effect on the mechanical properties [12,26]. In the case of a prosthesis requiring high rigidity, it is advantageous to fabricate the specimen in the same direction as the tensile direction. Additionally, it is considered important to understand the difference between yield strength and elongation according to the build direction, and to manufacture a clinically appropriate dental prosthesis in consideration of the characteristics of mechanical properties.

The 0.2% yield strength of the alloy can prevent permanent deformation of the restoration, especially in fixed partial denture frameworks, and a yield strength in excess of 300 MPa is sufficient to withstand permanent deformation [24]. All of the 3D-printing groups were 500 MPa or more, suitable for use in type 5.

Elongation refers to the physical deformation of a material before fracture when subjected to tensile stress and is related to the workability of the alloy [27]. Overall, the elongation was low in the 3D printing group, and a slight brittle property was also shown. Brittleness could not be overlooked even though the strength was much better compared to the other two groups. Despite this weakness, the ductility properties can be restored without significantly changing the microstructure through heat treatment, such as annealing [26,28]. In addition, when high-intensity ultrasonic waves are used, the microstructure of the metal can be converted from long columnar beta grain to equiaxed fine grain, thereby improving mechanical properties (yield strength and tensile strength) [29].

The tensile strain method was used to measure the modulus of elasticity in this study. In 2016, Ref. [20] introduced three methods of testing the modulus of elasticity for type-5 alloys (tensile deformation method, bending method in three- or four-point bending mode, and acoustic resonance method). The test was performed using the yield load value of 0.2% yield strength measured through the tensile test (Table 3). The modulus of elasticity was the highest in milling at 247 GPa, and all three groups met the requirement, which is 150 GPa. Previous studies have shown that the Co–Cr alloy used in the manufacture of dental prosthesis has a modulus of elasticity of 154–208 GPa [19]. All three groups are considered to have similar results. Modulus measurements were repeated at least four times and the average of the three most similar values was calculated. As a result of measuring the modulus of elasticity, there was no statistically significant difference between the three groups (*p* > 0.05). Numerical consistency was maintained and large standard deviations did not appear, increasing the accuracy of the results [23].

In the case of casting, since the fabrication of the prosthesis is performed manually, the result of the work varies depending on the skill level of the dental technician, so the technique of the dental technician is important. In the case of milling, the accuracy of the prosthesis varies according to the size of the bur used. A bur with a large size has good durability and excellent cutting rate, but a bur with a small diameter is useful for reproducing fine parts [1]. This precise cutting is one of the important factors in the CAD/CAM milling process. In 3D printing, it was found that a number of factors, such as the melting speed of the laser, the size of the metal powder, and the direction of the stacked build, affect the mechanical properties [26].

In addition, for the evaluation of dental materials, the concentration and dispersion of stresses applied to the restoration in the actual oral cavity should be considered [30]. Additionally, since the mechanical properties required by international standards must also be complied with, it is necessary to evaluate the standard test of dental materials.

In this study, the alloy composition of the material could not be completely identical, so it has a limitation in that more precise mechanical properties could not be compared.

Therefore, it is recommended that dental technicians and dentists who manufacture prostheses accurately understand the characteristics of the manufacturing method and select the appropriate method for clinical use. Advanced technology and digitalization in the dental field is expected to further increase the growth of the dental market.

This study was conducted on dental alloys, as the dental market has an increasing interest in esthetic dentistry. So, it will be interesting to conduct research on CAD/CAM dental materials, such as resins and ceramics, in the future. Additionally, some variables can change the results of mechanical tests. Surface abrasion or brushing can affect the mechanical properties and surface smoothness of dental materials [31,32]. It is considered a limitation in this study that there was no comparative study of mechanical properties considering surface treatment or clinical use to improve material performance. Therefore, further reports will be needed to carefully consider these materials and conditions.

## 5. Conclusions

The yield strength tended to be high in the 3D-printing group, especially at an angle of 0 degrees, where the tensile and stacking directions were horizontal, and the results were different depending on the build direction. The 3D printing group showed brittle properties due to low elongation compared to high yield strength, but this can be overcome by heat treatment, such as annealing. The results of the tensile strain method were numerically consistent, and the standard deviation was not great and the accuracy was high. All three groups satisfied the requirements of the international standard. Different structures between the three groups were observed through the image of the fracture surface of the specimen observed with the field emission scanning electron microscopes (FE-SEM). The structural difference is due to the manufacturing method, and it is believed that it affects the mechanical properties. Therefore, it is recommended to understand the characteristics of various manufacturing methods when fabricating dental prosthesis and to use it clinically.

## Figures and Tables

**Figure 1 materials-14-03367-f001:**
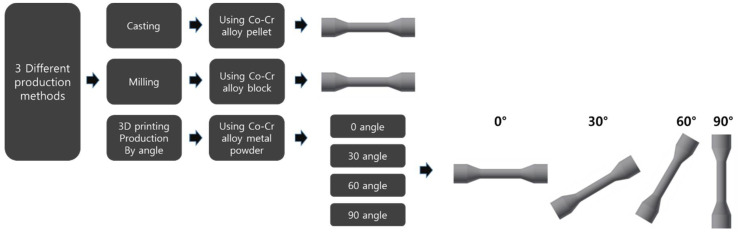
Three methods of manufacturing specimens for comparison of mechanical properties.

**Figure 2 materials-14-03367-f002:**
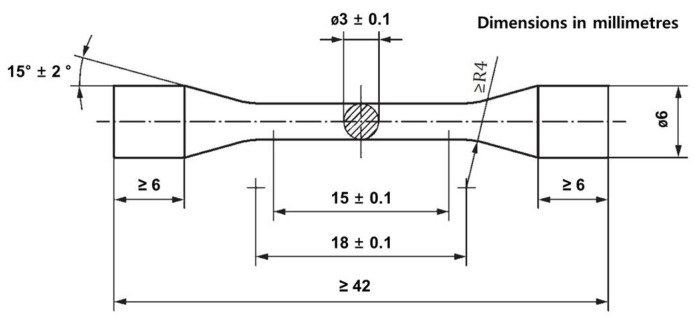
Dumbbell-shaped test specimens manufactured according to schematic drawing of ISO 22674.

**Figure 3 materials-14-03367-f003:**
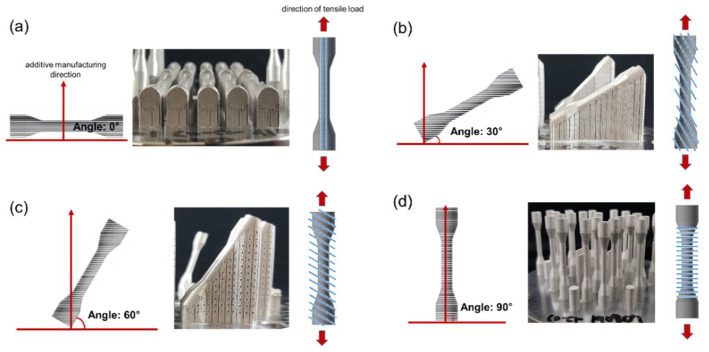
The additive manufacturing direction and the direction of tensile load: (**a**) 0°; (**b**) 30°; (**c**) 60°; (**d**) 90°.

**Figure 4 materials-14-03367-f004:**
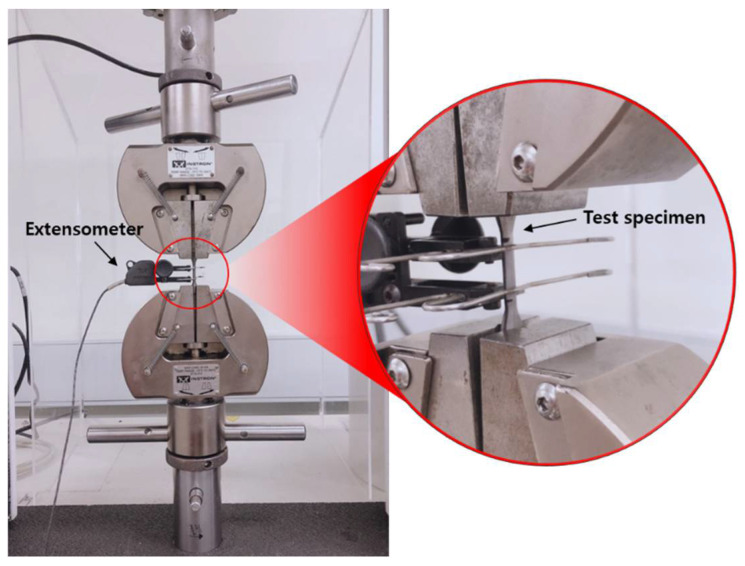
Tensile tests performed to evaluate mechanical properties.

**Figure 5 materials-14-03367-f005:**
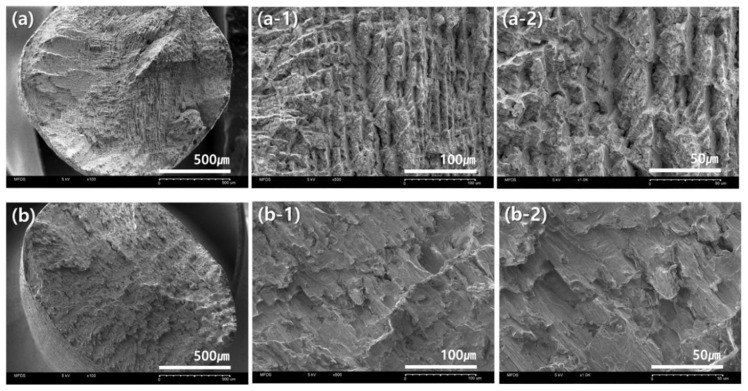
Observation of microstructure of fracture surface after tensile test: (**a**–**a-2**) casting specimen; (**b**–**b-2**) milling specimen. (**a**,**b**) ×100; (**a-1**,**b-1**) ×500; (**a-2**,**b-2**) ×1000.

**Figure 6 materials-14-03367-f006:**
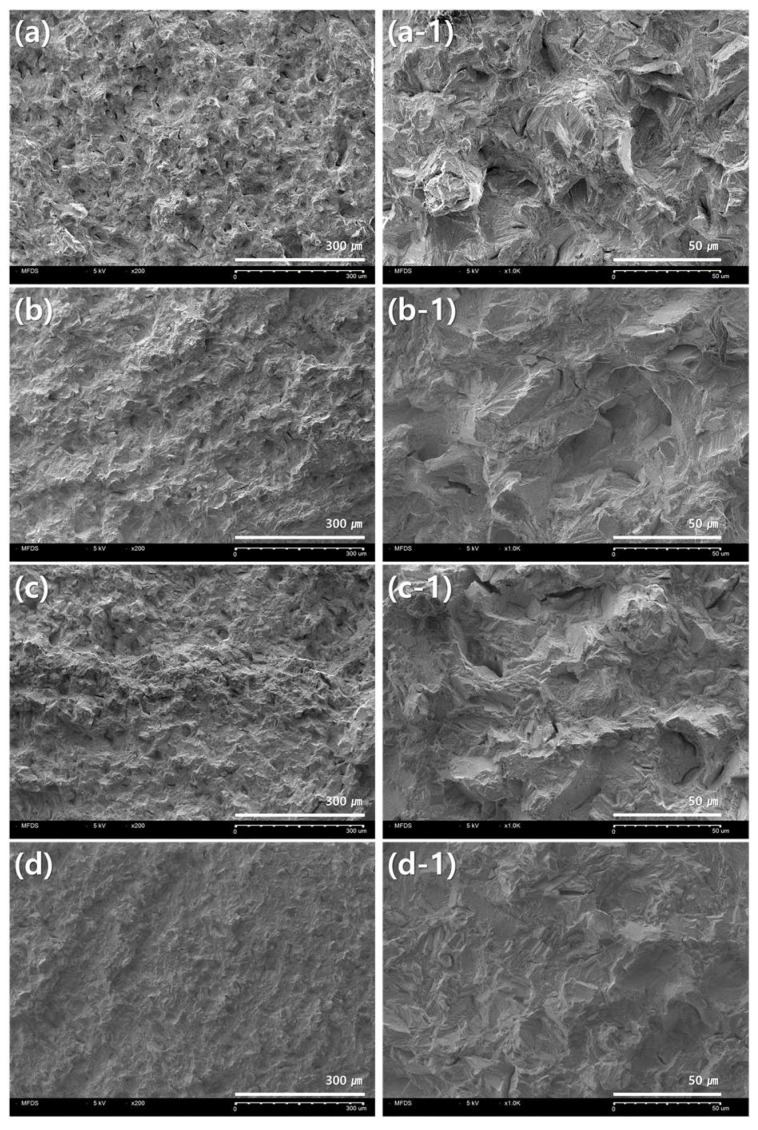
Observation of microstructure of fracture surface after tensile test of 3D printing group: (**a**,**a-1**) 0 degree; (**b**,**b-1**) 30 degree; (**c**,**c-1**) 60 degree; (**d**,**d-1**) 90 degree; (**a**–**d**) ×200; (**a-1**–**d-1**) ×1000.

**Table 1 materials-14-03367-t001:** Composition.

Group	Materials	Country	Composition	Manufacturer
Casting	Wirobond^®^C	Germany	Co: 63.3 wt.%, Cr: 24.8 wt.%, W: 5.3 wt.%, Mo: 5.1 wt.%, Si: 1 wt.%	Bego (Bremen)
Milling	Easymill	Korea	Co: >61 wt.%, Cr: >28 wt.%, Others	High dental (Gwang-ju)
3D printing	Cobalt alloys	Sweden	Co: Balance, Cr: 28.7 wt.%, Mo: 6.1 wt.%, Mn: 0.69 wt.%, Si: 0.68 wt.%, C: 0.25 wt.%, Fe: 0.18 wt.%, Ni: 0.01 wt.%	SANDVIK (Stockholm)

**Table 2 materials-14-03367-t002:** Evaluation of mechanical properties according to the specimen manufacturing method.

Group	0.2% Yield Strength (MPa)	Elongation (%)
	Mean ± Standard Deviation	Mean ± Standard Deviation
Casting	501 ± 33 ^a^	11 ± 3 ^AB^
Milling	438 ± 15 ^a^	12 ± 5 ^B^
3D printing *	1008 ± 59 *	9 ± 2 *
3D—0°	1042 ± 102 ^b^	5 ± 1 ^C^
3D—30°	1022 ± 86 ^b^	6 ± 1 ^AC^
3D—60°	1002 ± 32 ^b^	9 ± 2 ^ABC^
3D—90°	966 ± 47 ^b^	14 ± 2 ^B^

* 3D printing result means the average value. ^ab^ Different letters correspond to statistical differences for group (*p* < 0.05). ^ABC^ Different letters correspond to statistical differences for group (*p* < 0.05). Values a,b,c are subsets of the significance level of 0.05, and the mean difference for each group is significant at the level of 0.05 (*p* < 0.05).

**Table 3 materials-14-03367-t003:** Elastic modulus result value of tensile strain method measured through 0.2% yield strength.

Group	Yield Load (100%)	Yield Load (60%)	Yield Load (5%)	Elastic Modulus (GPa)	*p*-Value
Casting	3500.45	2100.27	175.02	226	*p* > 0.05
Milling	3152.41	1891.45	157.62	235	
3D—0°	6905.63	4143.38	345.28	219	
3D—30°	6760.25	4056.15	338.01	212	
3D—60°	6617.40	3970.44	330.87	206	
3D—90°	6387.27	3832.36	319.36	198	

The results of the elastic modulus measurement did not show any significant difference between the groups (*p* > 0.05, α = 0.394).

**Table 4 materials-14-03367-t004:** Mechanical properties by type specified in ISO 22674.

Type	0.2% Yield Strength (MPa)	Elongation (%)	Elastic Modulus (GPa)
0	-	-	-
1	80	18	-
2	180	10	-
3	270	5	-
4	360	2	-
5	500	2	150

## Data Availability

Not applicable.

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
