# Peer review of "Mechanical Properties of Dental Alloys According to Manufacturing Process"

_materials, 2021, doi:10.3390/ma14123367_

Round 1

Reviewer 1 Report

This study investigated the mechanical properties of Co-Cr alloys fabricated by casting, milling, printing methods. The results that the fabrication method affects the mechanical properties. I recommended the paper after some revisions.

  1. Specimen production

In lines 92–128, the preparation process is described. Please add the surface polishing method for each alloy. The surface state of the specimen affects on the mechanical properties.

  1. Table 2 and Fig. 5

Data is duplicated. The table and figure should be consolidated.

  1. The results of Fig. 7

In lines 224–226, “When the metal powder was melted by irradiation with a laser, it is believed that a gap formed due to incomplete dissolution of the power was observed.”

Is this speculation? If yes, this sentence should move to discussion section.

  1. Composition of Co-Cr alloys

In lines 256–260, the author discussed effect of the composition on the mechanical properties. Which group has the highest mechanical properties, judging from the composition? Please discuss.

  1. Crystal phase in each alloy

In lines 261–268, the author discussed effect of crystal phase for each alloy. However, this is speculation. The XRD measurement should be carried out to identify crystal phases in the Co-Cr alloys.

  1. Printed layers

In Fig. 7, we cannot observe layer structure. I concern why there is no layers in the specimen. If possible, please give the reason.

Author Response

Dear Managing Editor
Thank you for inviting us to submit a revised draft of our manuscript entitled, “Mechanical properties of dental alloys according to manufacturing process” to materials. 
We also appreciate the time and effort you and each of the reviewers have dedicated to providing insightful feedback on ways to strengthen our paper. 
Thus, it is with great pleasure that we resubmit our article for further consideration. We have incorporated changes that reflect the detailed suggestions you have graciously provided. We also hope that our edits and the responses we provide below satisfactorily address all the issues and concerns you and the reviewers have noted.
To facilitate your review of our revisions, the following is a point-by-point response to the questions and comments delivered in your letter.

1. Specimen production
In lines 92–128, the preparation process is described. Please add the surface polishing method for each alloy. The surface state of the specimen affects on the mechanical properties.
[Response] Thank you for your suggestion. However the surface treatment process of each materials was already described after the introduction of each material. Casted speciemens, milled speciemens and 3D printing speciemens were described respectively In the line 112-113 ”The casted specimens were cooled to room temperature and cleaned by sandblasting with alumina particles (50㎛)”, 117-118” . The milled specimen was cleaned by sandblasting with alumina particles (50㎛)”, 131-132 “All the 3D printing specimen was cleaned by sandblasting with alumina particles (50㎛)”.

2. Table 2 and Fig. 5
Data is duplicated. The table and figure should be consolidated.
[Response] thank you for checking my error and I agree with you. Thus, Table 2 and Figure 5 are consolidated deleting Figure 5.

3. The results of Fig. 7
In lines 224–226, “When the metal powder was melted by irradiation with a laser, it is believed that a gap formed due to incomplete dissolution of the power was observed.”
Is this speculation? If yes, this sentence should move to discussion section.
[Response] Thank you for good comment and we agree with you. Lines 227-229 were moved into the discussion section.

4. Composition of Co-Cr alloys
In lines 256–260, the author discussed effect of the composition on the mechanical properties. Which group has the highest mechanical properties, judging from the composition? Please discuss.
[Response] You have raised an important question. In the case of 3D printed specimens, the yield strength was overwhelmingly superior to that of other groups, but it is difficult to judge that the mechanical properties are the highest because the elongation is low and brittle. The Co-Cr alloy used in the test is composed of various elements such as Mo, W, Ni, and Si, and the content of each element is affected by strength and corrosion resistance, but since the test specimen was manufactured in three ways, the composition It is somewhat difficult to judge the mechanical properties by itself. It is believed that the mechanical properties are affected by complex interactions such as the components that make up the alloy and the way the prosthesis is manufactured. And added in lines 260-267.

5. Crystal phase in each alloy
In lines 261–268, the author discussed effect of crystal phase for each alloy. However, this is speculation. The XRD measurement should be carried out to identify crystal phases in the Co-Cr alloys.
[Response] Thank you for your suggestion. Although the XRD measurement was not performed in this study, a review of the previous study (reference No. 24) confirmed the data comparing the XRD of the casting specimen and the 3D printed specimen. It was confirmed that the α-phase is more dominant in the 3D printing group than casting group.

6. Printed layers
In Fig. 7, we cannot observe layer structure. I concern why there is no layers in the specimen. If possible, please give the reason.
[Response] Thank you for your suggestion. Since this test specimen was not subjected to additional surface coating treatment, the layer structure was not observed, and the surface was observed only by taking an SEM image.

Reviewer 2 Report

1) Please place keywords in alphabetic order.
2) The abstract should be shortened in lines 12-18, 30-33.
3) The particularity of this work should be emphasized in the abstract. There are plenty of articles devoted to the properties of known alloy compared to other production methods and the specimen orientation angle. Only in 2021, there are published 12 articles devoted to a similar subject.
4) The novelty of the work and study's tasks should be pronounced at the end of the introduction.
5) Composition should be presented uniformly in wt.%.
6) Please provide casting equipment, milling modes for 2 groups of specimens; how did you choose SLS modes? Did you conduct the experiments with varying modes? How did you decide that this laser scanning mode provides the optimum mechanical properties?
7) How many specimens were produced for each type of production?
8) I would not separate a single paragraph in subchapters such as 2.3, 2.4. Otherwise, I would devote a subchapter for each production method, for materials (how did you control powder granulometry, and how did you pre-treat the powder (drying, sieving), have you done post-treatment of SLS specimens to reduce anisotropy?) 
9) What do you think about the role of milling on tensile strength if you produce your milling specimens from the cast workpiece? They can also be produced from the SLS workpiece. The difference in mechanical properties between the cast, milled, and laser additively manufactured specimens is not correct since milling specimens are also made from cast blanks.
10) Figure 5 is not readable. 
11) Please add errors in Tables 2,3.
12) "due to incomplete dissolution of the power " - what does it mean? How can power be dissolved? Please use corrected terminology.
13) Lines 228-233 have no relation to the discussion. They are suitable for the abstract and introduction. Usually, the reader knows about your study by the discussion section, and it is not necessary to pronounce it once again.
14) If you did not measure roughness, do not talk about a rough surface.
15) What was the porosity of each sample?
16) Actually, it is not correct to compare various chemical compositions of the specimens if you aim to compare production methods.
17) "Co-Cr alloy undergoes a phase transition to α phase (FCC) at high temperature and ε (HCP) phase at low temperature" - it was never proved in the article.
18) The most important thing that was not mentioned in the discussion - what should be the properties to ensure the normal functioning of the prosthesis. It has long been known that a prosthesis should not be stronger and more rigid than a natural tooth since, during regular use (chewing), it can destroy and crumble natural teeth that work with a prosthesis in a pair of friction, in contact. Otherwise, projection is meaningless.
19) In the discussion, work should be carried out to compare the results of this work with the requirements for the operation of implants and other works devoted to studying the properties of the material for prosthetics.
20) The conclusions should be fundamental and have a further outlook.

Author Response

Dear Managing Editor
Thank you for inviting us to submit a revised draft of our manuscript entitled, “Mechanical properties of dental alloys according to manufacturing process” to materials. 
We also appreciate the time and effort you and each of the reviewers have dedicated to providing insightful feedback on ways to strengthen our paper. 
Thus, it is with great pleasure that we resubmit our article for further consideration. We have incorporated changes that reflect the detailed suggestions you have graciously provided. We also hope that our edits and the responses we provide below satisfactorily address all the issues and concerns you and the reviewers have noted.
To facilitate your review of our revisions, the following is a point-by-point response to the questions and comments delivered in your letter.

1) Please place keywords in alphabetic order.
[Response] Thank you for your suggestion. The keywords have been modified in alphabetical order.

2) The abstract should be shortened in lines 12-18, 30-33.
[Response] Thank you for your suggestion. This section has been summarized and revised.

3) The particularity of this work should be emphasized in the abstract. There are plenty of articles devoted to the properties of known alloy compared to other production methods and the specimen orientation angle. Only in 2021, there are published 12 articles devoted to a similar subject.
[Response] Thank you for your suggestion. The purpose of this study is to investigate the effect of the fabrication method of dental prosthesis on the mechanical properties as described in lines 8-9.
Corrected and supplemented the sentences in lines 8-9.

4) The novelty of the work and study's tasks should be pronounced at the end of the introduction.
[Response] We agree with you and have incorporated this suggestion throughout our paper. Added line 33-35 sentence.

5) Composition should be presented uniformly in wt.%.
[Response] Thank you for your suggestion. It has been modified to wt.% of Table 2.

6) Please provide casting equipment, milling modes for 2 groups of specimens; how did you choose SLS modes? Did you conduct the experiments with varying modes? How did you decide that this laser scanning mode provides the optimum mechanical properties?
[Response] You have raised an important question. Casting uses casting machines to melt the alloy. For milling mode, pistis' hyperdent cam software program was used. Added to lines 112 and 115. SLS production used the default set-up mode as described in lines 125-127. Since 3D printing technology has various variables such as laser scan speed and type of metal powder, you need to experiment yourself to see which musk provides the optimal conditions.

7) How many specimens were produced for each type of production?
[Response] Thank you for your suggestion. 6 specimens were used for the tensile strength test and 3 specimens were used for the modulus of elasticity test. Added to lines 136 and 158.

8) I would not separate a single paragraph in subchapters such as 2.3, 2.4. Otherwise, I would devote a subchapter for each production method, for materials (how did you control powder granulometry, and how did you pre-treat the powder (drying, sieving), have you done post-treatment of SLS specimens to reduce anisotropy?) 
[Response] Thank you for providing these insights. Combined 2.3 and 2.4 in line 163. As for the powder, a pre-treatment was not performed because the product that had already been processed provided by winforsys(manufacturer) was used. All the 3D printing specimen was cleaned by sandblasting with alumina particles (50㎛). And added to lines 131 and 132.

9) What do you think about the role of milling on tensile strength if you produce your milling specimens from the cast workpiece? They can also be produced from the SLS workpiece. The difference in mechanical properties between the cast, milled, and laser additively manufactured specimens is not correct since milling specimens are also made from cast blanks.
[Response] Thank you for your suggestion. It is true that the metal blocks used for milling are also made of cast blanks. However, the manufacturing process is considered to be more precise because the metal pellet is manually processed by a dental technician, and the metal block is cast by a machine through the manufacturing process. Therefore, the shape of the material is the same, but the processing process is different, so it is believed that it will affect the difference in mechanical properties.
10) Figure 5 is not readable. 
[Response] We agree with you and have incorporated this suggestion throughout our paper. Table 2 and Figure 5 are duplicates, so we will delete Figure 5.

11) Please add errors in Tables 2,3.
[Response] Thank you for your suggestion. Lines 189-196 and 216 added the p-value and significance level.

12) "due to incomplete dissolution of the power " - what does it mean? How can power be dissolved? Please use corrected terminology.
[Response] We agree with you. Lines 224-226 are also included in the discussion section and deleted.

13) Lines 228-233 have no relation to the discussion. They are suitable for the abstract and introduction. Usually, the reader knows about your study by the discussion section, and it is not necessary to pronounce it once again.
[Response] We agree with you. Lines 240-244 deleted.

14) If you did not measure roughness, do not talk about a rough surface.
[Response] Thank you for your suggestion. It is a word used to describe a surface that is relatively rougher than that of a cast specimen. Corrected "A rough surface" in line 263 to "an uneven surface".

15) What was the porosity of each sample?
[Response] Thank you for providing these insights. In the case of the casting and milling group, the porosity is partially observed at the fracture surface, and in the case of the specimen in the 3D printing group, it is difficult to measure the porosity due to the observation of the gap. If you look at (a-2) and (b-2) in Fig. 6, which are the photographs observed by magnifying the cross section of the test piece, the pore size is 10~20 ㎛, which is very small. 

16) Actually, it is not correct to compare various chemical compositions of the specimens if you aim to compare production methods.
[Response] You have raised an important question. The purpose of this study is to compare the mechanical properties according to the prosthesis manufacturing process. However, since the chemical composition of the raw materials of the test specimens of the 3 groups is not the same, so it is affected by complex interactions such as the composition of the alloy and the manufacturing method of the prosthesis.

17) "Co-Cr alloy undergoes a phase transition to α phase (FCC) at high temperature and ε (HCP) phase at low temperature" - it was never proved in the article.
[Response] You have raised an important question. According to the previous study (reference No. 24), the phase of Co-Cr alloy is converted into α phase (FCC) at high temperature and ε (HCP) phase at low temperature. It was possible to confirm the data comparing the XRD of the casting specimen and the 3D printed specimen. It was confirmed that the α phase is dominant in the 3D printed specimen than the casting, and a reference was added.

18) The most important thing that was not mentioned in the discussion - what should be the properties to ensure the normal functioning of the prosthesis. It has long been known that a prosthesis should not be stronger and more rigid than a natural tooth since, during regular use (chewing), it can destroy and crumble natural teeth that work with a prosthesis in a pair of friction, in contact. Otherwise, projection is meaningless.
[Response] You have raised an important question. Like your opinion, it is not a good prosthesis because it is unconditionally strong and hard. For long-term use while maximizing the function of the prosthesis in the oral cavity, it is necessary to make elaborate and delicate fabrication while satisfying the mechanical properties required by each type according to the use as shown in Table 4. Since the existing casting technique is made by hand, the result is variable depending on the technician's skill level, and the suitability of the restoration cannot be adjusted. It is thought that it is necessary to introduce a variety of digital technologies to the dental market to ensure the function of the prosthesis.

19) In the discussion, work should be carried out to compare the results of this work with the requirements for the operation of implants and other works devoted to studying the properties of the material for prosthetics.
[Response] Thank you for your suggestion. The introduction of digital technologies such as CAD/CAM into the dental market is easy for custom products such as removable dentures and implants, and results in excellent clinical results. Thus, clinicians can come to practical and economical conclusions due to their expertise and high-tech working methods. The information has been added to lines 333-337.

20) The conclusions should be fundamental and have a further outlook.
[Response] Thank you for your suggestion. The introduction of various advanced technologies and digital equipment is expected to bring high prospects for the growth of the dental market. The information has been added to lines 340-341.

Reviewer 3 Report

Dear Authors,

I have read the manuscript with interest and some questions raised. Enlisted please find my comments.

Overall. General English grammar revision (Minor spelling errors).

Key words. “dentistry” and “dental alloy” could be added in my opinion.

Abstract. Please add the names of the statistical tests in this section.

Introduction. Authors stated “For the fabrication of dental prostheses, precision casting has long been used. This is a method of dissolving a metal in a mold obtained by carving and embedding a wax pattern and injecting it to complete a cast body, which is also called a wax loss method or an investment method”. Please add a reference for this statement.

Introduction. Authors stated “ZZZ”. Please add a reference for this statement.

Introduction. Authors stated “ZZZ”. Please add a reference for this statement.

Materials and Methods. Authors stated “All specimens were made of Co-Cr alloy, and were manufactured in the form of 98 dumbbell-shaped tensile specimens according to ISO 22674(Fig.2)”. Please add the number of specimens in each group. Please add if and how sample size calculation has been performed.

Materials and Methods. Authors stated “A CAD/CAM program was used in the milling process”. Please add details about program name, version, manufacturer, City and State.

Materials and Methods. Authors stated “In the CAM process, a dumb bell-shaped specimen was designed according to the drawing in Fig. 2 using 3D data, and an STL file was created and then transferred to a milling machine (PM5-All, Pistis, Korea) to delete the metal block for processing”. Please add city of the manufacturer.

Materials and Methods. Authors stated “Specimens prepared using a 3D printer were printed by selective laser sintering (SLS)”. Please add details about commercial name manufacturer, City and State.

Materials and Methods. Authors stated “A tensile load was ap-136 plied to the test specimen using a universal material tester (Instron 3367, Instron Co., U.S.A.) at a cross-head speed of 1.5 ± 0.5 mm/min until the specimen broke”. Please add a reference for this method.

Materials and Methods. Authors stated “After the tensile test, the cross section of the test specimen was observed using a scan-161 ning electron microscope (SNE 4500M, Scanning electron microscope, SEM, SEC CO., 162 LTD., Korea)”. Please add magnification

Statistics. Authors used ANOVA. ANOVA is used for gaussian distributions. Please explain how normality of data was tested.

Results. Please add P values in the main text all along this section.

Results. Figure 5. Please enlarge a bit the graphs in order to increase readability

Discussion. Authors could add some discussion about the other research perspectives and the clinical variables that could alter the results of the present report. It could be added that “The present report analyzed dental alloys. It would be interesting in the future to test also other CAD CAM dental materials such as resins and ceramics. Additionally some variables could alter the results of mechanical tests. In fact wear (Hardness and Wear Resistance of Dental Biomedical Nanomaterials in a Humid Environment with Non-Stationary Temperatures. Pieniak D, Walczak A, Walczak M, Przystupa K, Niewczas AM. Materials (Basel). 2020 Mar 10;13(5):1255) or tooth brushing (Effect of Long-Term Brushing on Deflection, Maximum Load, and Wear of Stainless Steel Wires and Conventional and Spot Bonded Fiber-Reinforced Composites. Scribante A, Vallittu P, Lassila LVJ, Viola A, Tessera P, Gandini P, Sfondrini MF. Int J Mol Sci. 2019 Nov 30;20(23):6043) can alter both mechanical characteristics and surface smoothness of dental materials. Therefore further reports are needed in order to take into careful account also these materials and conditions.”

Discussion. Please add a paragraph showing the limitations of the present report.

Discussion. Provide a general interpretation of the results in the context of other evidence, and implications for future research.

Conclusion. Please shorten a bit this section.

Author Response

Dear Managing Editor
Thank you for inviting us to submit a revised draft of our manuscript entitled, “Mechanical properties of dental alloys according to manufacturing process” to materials. 
We also appreciate the time and effort you and each of the reviewers have dedicated to providing insightful feedback on ways to strengthen our paper. 
Thus, it is with great pleasure that we resubmit our article for further consideration. We have incorporated changes that reflect the detailed suggestions you have graciously provided. We also hope that our edits and the responses we provide below satisfactorily address all the issues and concerns you and the reviewers have noted.
To facilitate your review of our revisions, the following is a point-by-point response to the questions and comments delivered in your letter.

1. Key words. “dentistry” and “dental alloy” could be added in my opinion.
[Response] Thank you for your suggestion. I added a keyword on line 34.

2. Abstract. Please add the names of the statistical tests in this section.
[Response] Thank you for your suggestion. Added the statistical test name to the abstract on lines 22-23.
*lines 22-23: statistics of the data were analyzed with a statistical program (IBM SPSS Statistics 27) and  using Anova and multiple comparison post tests.

3. Introduction. Authors stated “For the fabrication of dental prostheses, precision casting has long been used. This is a method of dissolving a metal in a mold obtained by carving and embedding a wax pattern and injecting it to complete a cast body, which is also called a wax loss method or an investment method”. Please add a reference for this statement.
[Response] Thank you for your suggestion. Added references on line 40.

4-5. Introduction. Authors stated “ZZZ”. Please add a reference for this statement.
[Response] Thank you for your suggestion. However, If you tell me again which sentence you are referring to, I will try to supplement it. 

6. Materials and Methods. Authors stated “All specimens were made of Co-Cr alloy, and were manufactured in the form of dumbbell-shaped tensile specimens according to ISO 22674(Fig.2)”. Please add the number of specimens in each group. Please add if and how sample size calculation has been performed.
[Response] You have raised an important question. It was added on lines 98-99 that the total number of specimens and the dimensions of the specimens were manufactured according to the drawings in Fig. 2.
*lines 98-99: All specimens were made of Co-Cr alloy, and the dimensions of the specimens were manufactured in the form of 54  dumbbell-shaped tensile specimens according to ISO 22674(Fig.2).

7. Materials and Methods. Authors stated “A CAD/CAM program was used in the milling process”. Please add details about program name, version, manufacturer, City and State.
[Response] Thank you for your suggestion. Added CAM process information on line 114 
*line 114: A CAD/CAM program(hyperdent cam software, pistis, korea, incheon)  was used in the milling process.

8. Materials and Methods. Authors stated “In the CAM process, a dumb bell-shaped specimen was designed according to the drawing in Fig. 2 using 3D data, and an STL file was created and then transferred to a milling machine (PM5-All, Pistis, Korea) to delete the metal block for processing”. Please add city of the manufacturer.
[Response] Thank you for your suggestion. Added manufacturer's city on line 117.
* line 117: milling machine (PM5-All, Pistis, Korea, incheon )

9. Materials and Methods. Authors stated “Specimens prepared using a 3D printer were printed by selective laser sintering (SLS)”. Please add details about commercial name manufacturer, City and State.
[Response] Thank you for your suggestion. Added information of commercial name manufacturer on lines 119-120.
* lines 119-120: 3D printer(METALSYS 120D, Winforsys, Korea, Gyeonggi-do)

10. Materials and Methods. Authors stated “A tensile load was ap-136 plied to the test specimen using a universal material tester (Instron 3367, Instron Co., U.S.A.) at a cross-head speed of 1.5 ± 0.5 mm/min until the specimen broke”. Please add a reference for this method.
[Response] Thank you for your suggestion. Added a reference on lines 141-142.
* lines 141-142: This test was performed according to ISO 22674-8.3.2 mechanical test procedure.

11. Materials and Methods. Authors stated “After the tensile test, the cross section of the test specimen was observed using a scan-161 ning electron microscope (SNE 4500M, Scanning electron microscope, SEM, SEC CO., 162 LTD., Korea)”. Please add magnification
[Response] Thank you for your suggestion. Added magnification on lines 165-166.
* lines 165-166: After the tensile test, the cross section of the test specimen was observed at 100 mag-nification, 200 magnification, 500 magnification, and 1000 magnification,  using a scanning electron microscope (SNE 4500M, Scanning electron microscope, SEM, SEC CO., LTD., Korea).

12. Statistics. Authors used ANOVA. ANOVA is used for gaussian distributions. Please explain how normality of data was tested.
[Response] Thank you for your suggestion. One-way ANOVA analysis was used to confirm the homogeneity of variance, and the Scheffe method was used for post-mortem analysis.

13. Results. Please add P values in the main text all along this section.
[Response] Thank you for your suggestion. Lines 190-197 and 218 added the p-value and significance level.
* Lines 190-197: 0.2 % yield strength: Casting and milling group p>0.05 (α=0.593), 3D printing groups(0, 30, 60, 90) p>0.05 (α=0.195). The casting and 3D printing groups(0, 30, 60, 90), and the milling and 3D printing groups(0, 30, 60, 90) showed significant differences as p<0.05. Elongation: casting and 3D30 and 3D60 p>0.05 (α=0.159). Casting and milling and 3D60 and 3D90 p>0.05 (α=0.187). 3D0 and 30 and 60 p>0.05 (α=0.111). The milling and 3D 30, the 3D 0 and 90, the 3D 30 and 90 showed significant differences as p<0.05.
* Lines 218: The results of the elastic modulus measurement did not show any significant difference between the groups(p > 0.05, α=0.394).
14. Results. Figure 5. Please enlarge a bit the graphs in order to increase readability
[Response] We agree with you and have incorporated this suggestion throughout our paper. Table 2 and Figure 5 are duplicates, so we deleted Figure 5.

15. Discussion. Authors could add some discussion about the other research perspectives and the clinical variables that could alter the results of the present report. It could be added that “The present report analyzed dental alloys. It would be interesting in the future to test also other CAD CAM dental materials such as resins and ceramics. Additionally some variables could alter the results of mechanical tests. In fact wear (Hardness and Wear Resistance of Dental Biomedical Nanomaterials in a Humid Environment with Non-Stationary Temperatures. Pieniak D, Walczak A, Walczak M, Przystupa K, Niewczas AM. Materials (Basel). 2020 Mar 10;13(5):1255) or tooth brushing (Effect of Long-Term Brushing on Deflection, Maximum Load, and Wear of Stainless Steel Wires and Conventional and Spot Bonded Fiber-Reinforced Composites. Scribante A, Vallittu P, Lassila LVJ, Viola A, Tessera P, Gandini P, Sfondrini MF. Int J Mol Sci. 2019 Nov 30;20(23):6043) can alter both mechanical characteristics and surface smoothness of dental materials. Therefore further reports are needed in order to take into careful account also these materials and conditions.”
[Response] Thank you for your suggestion. The clinical characteristics of clinical variables will be carried out in future research projects.

16. Discussion. Please add a paragraph showing the limitations of the present report.
[Response] Thank you for providing these insights. In this study, the alloy composition of the material could not be completely identical, so it has a limitation that more precise mechanical properties could not be compared. Added on lines 340-342.

17. Discussion. Provide a general interpretation of the results in the context of other evidence, and implications for future research
[Response] Thank you for your suggestion. From these results, it is believed that advantageous results can be produced using various variables such as the size of the powder, the speed of the laser, the specifications of the 3D printing equipment, and the manufacturing angle. The introduction of various advanced technologies and digital equipment is expected to bring high prospects for the growth of the dental market. Added on lines 293-297.

18. Conclusion. Please shorten a bit this section.
[Response] Thank you for your suggestion. To short a bit this section, deleted unnecessary sentences on lines 345-346.

Reviewer 4 Report

The manuscript “Mechanical properties of dental alloys according to manufacturing process” describes the influence of fabrication method of dental prosthesis on the mechanical properties. The manuscript seem interesting. I think it should be specified that the study concerns the metallic materials used in fabrication of dental prostheses.

Please, correct the manuscript according editorial requirements: the abstract must be shorten to 200 words; correct the referencing style etc.

The introduction should be revised as containing inaccurate statements/information and reorganized. Please, sort out the advantages and disadvantages of described manufacturing methods.

Line 38: casting is not “dissolving a metal in a mold”, but melting the metal and pouring it into the mold. The whole sentence (lines 37-38) is too long, split in two and reorder information within it.

Explain abbreviations at their first use (line 45: CAD/CAM; line 51: STL)

Too long sentences (e.g. lines 52-55)

Some statements seem out of context, e.g. line 64 “There are methods such as heat treatment after processing”

Line 76: “excellent biocompatibility” of Ni-Cr alloys?

Line 80: “The mechanical strength of a dental prosthesis is closely related to the masticatory pressure” ?

Correct the description of Table 1. What does group 1, 2, 3 refer to? They are not mentioned further in the study. What was the control group? How many samples for each group was prepared? Line 309 mentions “Modulus measurements were repeated at least 4 times and the average of the 3 most similar values was calculated”

Table 2: Correct the “Values a,b,c..” in the table, e.g. use small and capital letters for yield strength and elongation results, respectively.  

Lines 175-176: the repetition of the information from the Table 2

Line 177: “The yield strength was higher in the horizontal than in the vertical build direction” is not precise

Lines 190-191: provide level of significance

Correct the description of Figure 5: a, b, c. Provide Figures with higher resolution

On Fig 5c what does p=0.394 refer to?

Fig 6. Correct the description : do images (a), (a-1), (a-2) differ only by the magnification, is it the image of the same sample?

Line 215: the roughness wasn’t measured in the study

Line 237: “The range of application is different for each type, and the highest type can include the low type, and the highest type, type 6,..” ???

Correct spelling mistakes e.g. table 2 (“yield”), double commas (line 44), additional spacing

Author Response

Dear Managing Editor
Thank you for inviting us to submit a revised draft of our manuscript entitled, “Mechanical properties of dental alloys according to manufacturing process” to materials. 
We also appreciate the time and effort you and each of the reviewers have dedicated to providing insightful feedback on ways to strengthen our paper. 
Thus, it is with great pleasure that we resubmit our article for further consideration. We have incorporated changes that reflect the detailed suggestions you have graciously provided. We also hope that our edits and the responses we provide below satisfactorily address all the issues and concerns you and the reviewers have noted.
To facilitate your review of our revisions, the following is a point-by-point response to the questions and comments delivered in your letter.

1. The manuscript “Mechanical properties of dental alloys according to manufacturing process” describes the influence of fabrication method of dental prosthesis on the mechanical properties. The manuscript seem interesting. I think it should be specified that the study concerns the metallic materials used in fabrication of dental prostheses.
[Response] Thank you for your suggestion. In lines 8-9, it describes the effect of the manufacturing method of the dental prosthesis on the mechanical properties, and has been slightly modified.

2. Please, correct the manuscript according editorial requirements: the abstract must be shorten to 200 words; correct the referencing style etc.
[Response] Thank you for your suggestion. The abstract was revised according to the editorial requirements.

3. The introduction should be revised as containing inaccurate statements/information and reorganized. Please, sort out the advantages and disadvantages of described manufacturing methods.
[Response] Thank you for your suggestion. The advantage of casting is the traditional manufacturing method that has been used for a long time, and the disadvantage is that it takes a long time. The advantage of milling is that it is easier to manufacture than the casting method and the result can be accurately controlled. Disadvantages are that a lot of raw materials for the metal block to be cut are consumed, the milling tool is frequently worn, and detailed work may be difficult. The disadvantage of SLS is that it has brittleness and is expensive. It is additionally described in line 74-75. The advantage is that the production speed is fast, the material waste is low, and the porosity generation rate is remarkably low.

4. Line 38: casting is not “dissolving a metal in a mold”, but melting the metal and pouring it into the mold. The whole sentence (lines 37-38) is too long, split in two and reorder information within it.
[Response] Thank you for your suggestion. Corrected as mentioned in lines 38-44.

5. Explain abbreviations at their first use (line 45: CAD/CAM; line 51: STL)
[Response] We agree with you. Added abbreviation of CAD/CAM in lines 50-51and STL in line 57

6.  Too long sentences (e.g. lines 52-55)
[Response] We agree with you. Separated sentences in line 60.

7. Some statements seem out of context, e.g. line 64 “There are methods such as heat treatment after processing”
 [Response] We agree with you. Deleted the sentence in line 70

8. Line 76: “excellent biocompatibility” of Ni-Cr alloys?
[Response] Thank you for your suggestion. Ni-Cr contents deleted in lines 81-82.

9.  Line 80: “The mechanical strength of a dental prosthesis is closely related to the masticatory pressure” ?
[Response] Thank you for your suggestion. If the prosthesis is harder than the tooth, it causes abrasion of the tooth, so it is expressed that it is related to the mechanical strength.

10. Correct the description of Table 1. What does group 1, 2, 3 refer to? They are not mentioned further in the study. What was the control group? How many samples for each group was prepared? Line 309 mentions “Modulus measurements were repeated at least 4 times and the average of the 3 most similar values was calculated”
[Response] You have raised an important question. Groups 1, 2, and 3 in Table 1 mean casting, milling, and 3D printing, and have been modified. The control group is the traditional casting method. The number of specimens is 6 tensile strength tests and 3 elastic modulus tests. It is described in line 140 and 162. When measuring the modulus of elasticity, one specimen was measured by repeating at least four times according to ISO 22674-8.6.1.2, and the average of the three most similar values was described.

11. Table 2: Correct the “Values a,b,c..” in the table, e.g. use small and capital letters for yield strength and elongation results, respectively. 
[Response] Thank you for your suggestion. Corrected spelling.

12. Lines 175-176: the repetition of the information from the Table 2
[Response] We agree with you. It is written to refer to the data in the table 2. However, as you mentioned, it can be checked in Table 2, so I deleted in lines 183-185.

13. Line 177: “The yield strength was higher in the horizontal than in the vertical build direction” is not precise
[Response] Thank you for your suggestion. This means that among the 3D printing groups, the yield strength of 0 degrees stacked horizontally was higher than 90 degrees stacked vertically. Lines 186-187 contains additional information.’

14. Lines 190-191: provide level of significance
[Response] Thank you for your suggestion. Added the p-value and significance level in lines 194-201.
15. Correct the description of Figure 5: a, b, c. Provide Figures with higher resolution
On Fig 5c what does p=0.394 refer to?
[Response] Thank you for your suggestion. Figure 5 and Table 2 are duplicates of data, so Figure 5 was deleted, and the significance level was added to line 221

16. Fig 6. Correct the description : do images (a), (a-1), (a-2) differ only by the magnification, is it the image of the same sample?
[Response] You have raised an important question. Same sample image, different magnification. Additional magnification information is provided in lines 168-169.

17.Line 215: the roughness wasn’t measured in the study
[Response] Thank you for your suggestion. Surface observation was performed through SEM images without measuring the roughness.

18. Line 237: “The range of application is different for each type, and the highest type can include the low type, and the highest type, type 6,..” ???
[Response] Thank you for your suggestion. This has been corrected in Lines 254-256.

19. Correct spelling mistakes e.g. table 2 (“yield”), double commas (line 44), additional spacing.
[Response] Thank you for your suggestion. Correct spelling.

Round 2

Reviewer 1 Report

Thank you for revision of your manuscript. I recommended this manuscript for acceptance after further revision as follow.

In your response,  "Thank you for your suggestion. Although the XRD measurement was not performed in this study, a review of the previous study (reference No. 24) confirmed the data comparing the XRD of the casting specimen and the 3D printed specimen. It was confirmed that the α-phase is more dominant in the 3D printing group than casting group."

The author should be clearly state above mentions in the discussion section. Crystal phase affect the mechanical properties. Therefore, the author should clarify it. 

Author Response

Dear Managing Editor
Thank you for inviting us to submit a revised draft of our manuscript entitled, “Mechanical properties of dental alloys according to manufacturing process” to materials. 
We also appreciate the time and effort you and each of the reviewers have dedicated to providing insightful feedback on ways to strengthen our paper. 
Thus, it is with great pleasure that we resubmit our article for further consideration. We have incorporated changes that reflect the detailed suggestions you have graciously provided. We also hope that our edits and the responses we provide below satisfactorily address all the issues and concerns you and the reviewers have noted.
To facilitate your review of our revisions, the following is a point-by-point response to the questions and comments delivered in your letter.

Comment 1. In your response,  "Thank you for your suggestion. Although the XRD measurement was not performed in this study, a review of the previous study (reference No. 24) confirmed the data comparing the XRD of the casting specimen and the 3D printed specimen. It was confirmed that the α-phase is more dominant in the 3D printing group than casting group."
The author should be clearly state above mentions in the discussion section. Crystal phase affect the mechanical properties. Therefore, the author should clarify it.

[Response] We agree with the reviewer's suggestion. We have revised the manuscript accordingly in lines 272-275. 

Reviewer 2 Report

Some of the reviewer's comments were ignored. The article can be published after the formulation of novelty at the end of the introduction, research objectives, revision of the conclusion following the previous remarks. The abstract, discussion, and conclusions should indicate that the metal cannot work in a pair of friction with a more brittle material. It is advised to provide the physical and mechanical properties of the tooth and filling to compare the studies' results with the characteristics of the materials in the oral cavity.

Author Response

Dear Managing Editor
Thank you for inviting us to submit a revised draft of our manuscript entitled, “Mechanical properties of dental alloys according to manufacturing process” to materials. 
We also appreciate the time and effort you and each of the reviewers have dedicated to providing insightful feedback on ways to strengthen our paper. 
Thus, it is with great pleasure that we resubmit our article for further consideration. We have incorporated changes that reflect the detailed suggestions you have graciously provided. We also hope that our edits and the responses we provide below satisfactorily address all the issues and concerns you and the reviewers have noted.
To facilitate your review of our revisions, the following is a point-by-point response to the questions and comments delivered in your letter.

1. Some of the reviewer's comments were ignored. The article can be published after the formulation of novelty at the end of the introduction, research objectives, revision of the conclusion following the previous remarks. The abstract, discussion, and conclusions should indicate that the metal cannot work in a pair of friction with a more brittle material. It is advised to provide the physical and mechanical properties of the tooth and filling to compare the studies' results with the characteristics of the materials in the oral cavity.

[Response] Thank you for your suggestion. First of all, we are very sorry that some of the reviewers' comments were not well reflected. We added novelty at the end of the introduction in lines 92-95. It was added to the manuscript in lines 250-253 that the wear of the two pairs should be similar in order for the metal to not wear out from friction with more brittle materials or natural teeth. Added novelty at the end of the introduction. It added in lines 346-349 that the results of this study were necessary for the evaluation of the properties of oral materials.

Reviewer 3 Report

Dear Authors,

The text has been revised but some major concerns still remain unsolved.

Enlisted please find some considerations.

Abstract. Authors stated “After the tensile test, the cross section of the fractured specimen was observed with a scanning elec-16 tron microscope, and the statistics of the data were analyzed with a statistical program (IBM SPSS 17 Statistics 27)”. Please avud commercial names (Software) in this section.

Abstract. Please add the names of the statistical post hoc tests in this section.

Materials and Methods. Authors stated “All specimens were made of Co-Cr alloy, and the dimensions of the specimens were 93 manufactured in the form of 54 dumbbell-shaped tensile specimens according to ISO 94 22674(Fig.2)”. Please add how sample size calculation has been performed.

Materials and Methods. Authors stated “A CAD/CAM program (hyperdent cam software, pistis, korea, Incheon) was used in 111 the milling process”. Please add details about version.

Materials and Methods. Authors stated “After the tensile test, the cross section of the test specimen was observed at 100 magnification, 200 magnification, 500 magnification, and 1000 magnification”. Please rephrase as “After the tensile test, the cross section of the test specimen was observed at x100, x200, x500, and x1000 magnification”.

Statistics. Authors used ANOVA. ANOVA is used for gaussian distributions. Please explain how normality of data was tested.

Results. Please add P values in the main text all along this section.

Results. Graphical representation of the results have been removed. Some graphs could help the reader in understanding the results of the report.

Discussion. Authors could add some discussion about the other research perspectives and the clinical variables that could alter the results of the present report. It could be added that “The present report analyzed dental alloys. It would be interesting in the future to test also other CAD CAM dental materials such as resins and ceramics. Additionally some variables could alter the results of mechanical tests. In fact wear (Hardness and Wear Resistance of Dental Biomedical Nanomaterials in a Humid Environment with Non-Stationary Temperatures. Pieniak D, Walczak A, Walczak M, Przystupa K, Niewczas AM. Materials (Basel). 2020 Mar 10;13(5):1255) or tooth brushing (Effect of Long-Term Brushing on Deflection, Maximum Load, and Wear of Stainless Steel Wires and Conventional and Spot Bonded Fiber-Reinforced Composites. Scribante A, Vallittu P, Lassila LVJ, Viola A, Tessera P, Gandini P, Sfondrini MF. Int J Mol Sci. 2019 Nov 30;20(23):6043) can alter both mechanical characteristics and surface smoothness of dental materials. Therefore further reports are needed in order to take into careful account also these materials and conditions.”

Discussion. Please add a paragraph showing the limitations of the present report.

Conclusion. Please shorten a bit this section.

Author Response

Dear Managing Editor
Thank you for inviting us to submit a revised draft of our manuscript entitled, “Mechanical properties of dental alloys according to manufacturing process” to materials. 
We also appreciate the time and effort you and each of the reviewers have dedicated to providing insightful feedback on ways to strengthen our paper. 
Thus, it is with great pleasure that we resubmit our article for further consideration. We have incorporated changes that reflect the detailed suggestions you have graciously provided. We also hope that our edits and the responses we provide below satisfactorily address all the issues and concerns you and the reviewers have noted.
To facilitate your review of our revisions, the following is a point-by-point response to the questions and comments delivered in your letter.

1. Abstract. Authors stated “After the tensile test, the cross section of the fractured specimen was observed with a scanning electron microscope, and the statistics of the data were analyzed with a statistical program (IBM SPSS 17 Statistics 27)”. Please avud commercial names (Software) in this section.[Response] Thank you for your suggestion. We added commercial names  of SPSS (IBM Corp. Released 2020. IBM SPSS Statistics for Windows, Version 27.0. Armonk, NY: IBM Corp).

2. Abstract. Please add the names of the statistical post hoc tests in this section.
[Response] Thank you for your suggestion. We added names of the statistical post hoc tests

3. Materials and Methods. Authors stated “All specimens were made of Co-Cr alloy, and the dimensions of the specimens were 93 manufactured in the form of 54 dumbbell-shaped tensile specimens according to ISO 94 22674(Fig.2)”. Please add how sample size calculation has been performed. 
[Response] Thank you for your suggestion. Specimens were drawn using AutoCAD software(Inventor ver.2015, Autodesk, Inc., California, USA) according to the dimensions of the in Figure 2, and then converted into .stl format and sent to the machine.

4. Materials and Methods. Authors stated “A CAD/CAM program (hyperdent cam software, pistis, korea, Incheon) was used in 111 the milling process”. Please add details about version.
[Response] Thank you for your suggestion. We added software details(hyperdent cam software v.8.2).

5. Materials and Methods. Authors stated “After the tensile test, the cross section of the test specimen was observed at 100 magnification, 200 magnification, 500 magnification, and 1000 magnification”. Please rephrase as “After the tensile test, the cross section of the test specimen was observed at x100, x200, x500, and x1000 magnification”.
[Response] We agree with you. We have edited that paragraph as you mentioned

6. Statistics. Authors used ANOVA. ANOVA is used for gaussian distributions. Please explain how normality of data was tested.
[Response] Thank you for your suggestion. For the normality test, shapiro-wilk was used, and it was confirmed that the samples of all groups were P > 0.05.

7. Results. Please add P values in the main text all along this section.
[Response] Thank you for your suggestion. We added p value.

8. Results. Graphical representation of the results have been removed. Some graphs could help the reader in understanding the results of the report.
[Response] Thank you for providing these insights. However, the graph (past figure 5) duplicated with contents of Table 2, and the description of the graph is written in the manuscript, so it was finally excluded.

9. Discussion. Authors could add some discussion about the other research perspectives and the clinical variables that could alter the results of the present report. It could be added that “The present report analyzed dental alloys. It would be interesting in the future to test also other CAD CAM dental materials such as resins and ceramics. Additionally some variables could alter the results of mechanical tests. In fact wear (Hardness and Wear Resistance of Dental Biomedical Nanomaterials in a Humid Environment with Non-Stationary Temperatures. Pieniak D, Walczak A, Walczak M, Przystupa K, Niewczas AM. Materials (Basel). 2020 Mar 10;13(5):1255) or tooth brushing (Effect of Long-Term Brushing on Deflection, Maximum Load, and Wear of Stainless Steel Wires and Conventional and Spot Bonded Fiber-Reinforced Composites. Scribante A, Vallittu P, Lassila LVJ, Viola A, Tessera P, Gandini P, Sfondrini MF. Int J Mol Sci. 2019 Nov 30;20(23):6043) can alter both mechanical characteristics and surface smoothness of dental materials. Therefore further reports are needed in order to take into careful account also these materials and conditions.”
[Response] Thank you for your suggestion. We added some discussion about the other research perspectives and the clinical variables in lines 350-357.

10. Discussion. Please add a paragraph showing the limitations of the present report.
[Response] Thank you for your suggestion. We added the limitations of the present report in lines 354-356.

11. Conclusion. Please shorten a bit this section.
[Response] Thank you for your suggestion. To short a bit this section, deleted unnecessary sentences on lines 363-365.

Reviewer 4 Report

Thank you for providing corrected version of your manuscript.

Minor issues:

Please, correct the referencing style.

The added sentences in lines 28-30:

“Attach a sprue pin to the wax pattern made as a model of the specimen and put it in the investment material. The sprue pin provides a passage for molten metal into the mold.”

seem a bit out of context. If they serve as explanation to casting technique, please connect them with previous sentence.

Some added statements highly need grammar and style correction.

Line 75: The statement “The mechanical strength of a dental prosthesis is closely related to the masticatory pressure” needs to be rewritten.

The Authors explanation (“If the prosthesis is harder than the tooth, it causes abrasion of the tooth, so it is expressed that it is related to the mechanical strength.”) proves the sentence is badly written and can be misinterpreted. What I understand from the statement is that the dental prosthesis is as strong as low occlusal load is applied.

Fig 6. If images (a), (a-1), (a-2) on Figure 6 (now Fig 5) differ only by the magnification, add the info about magnification in Figure description as well.

Line 215: If the roughness wasn’t measured in the study, please use other word than “rough”.

Author Response

Dear Managing Editor
Thank you for inviting us to submit a revised draft of our manuscript entitled, “Mechanical properties of dental alloys according to manufacturing process” to materials. 
We also appreciate the time and effort you and each of the reviewers have dedicated to providing insightful feedback on ways to strengthen our paper. 
Thus, it is with great pleasure that we resubmit our article for further consideration. We have incorporated changes that reflect the detailed suggestions you have graciously provided. We also hope that our edits and the responses we provide below satisfactorily address all the issues and concerns you and the reviewers have noted.
To facilitate your review of our revisions, the following is a point-by-point response to the questions and comments delivered in your letter.

1. Please, correct the referencing style.
[Response] We agree with the reviewer's suggestion. We have revised the manuscript accordingly. 

2. The added sentences in lines 28-30:
“Attach a sprue pin to the wax pattern made as a model of the specimen and put it in the investment material. The sprue pin provides a passage for molten metal into the mold.”
seem a bit out of context. If they serve as explanation to casting technique, please connect them with previous sentence.
[Response] Thank you for your suggestion. A new sentence has been added to connect with the previous sentence as an description of the casting technique.

3. Some added statements highly need grammar and style correction.
[Response] Thank you for your suggestion. We corrected grammar and sentence style.

4. Line 75: The statement “The mechanical strength of a dental prosthesis is closely related to the masticatory pressure” needs to be rewritten.
The Authors explanation (“If the prosthesis is harder than the tooth, it causes abrasion of the tooth, so it is expressed that it is related to the mechanical strength.”) proves the sentence is badly written and can be misinterpreted. What I understand from the statement is that the dental prosthesis is as strong as low occlusal load is applied.
[Response] We agree with the reviewer's suggestion. Part of the sentence has been deleted to avoid misunderstandings that If the prosthesis is harder than the tooth, it causes abrasion of the tooth, so it is expressed that it is related to the mechanical strength.

5. Fig 6. If images (a), (a-1), (a-2) on Figure 6 (now Fig 5) differ only by the magnification, add the info about magnification in Figure description as well.
[Response] We agree with the reviewer's suggestion. Added information about magnification to the Figure descriptions in Figures 5 and 6.

6. Line 215: If the roughness wasn’t measured in the study, please use other word than “rough”.
[Response] Thank you for your suggestion. It was a word used to describe a surface that was relatively rougher than the cast specimen, but revised the word "rough surface" in line 217 to "uneven surface"
